

# Regularization Destriping of Remote Sensing Imagery

Ranil Basnayake[1], Erik Bollt[1], Nicholas Tufillaro[2], Jie Sun[1], and Michelle Gierach[3]

[1]Department of Mathematics, Clarkson University, 8 Clarkson Avenue 5815, Potsdam, NY 13699, USA.
[2]College of Earth, Ocean, and Atmospheric Sciences, Oregon State University, 104 CEOAS Administration
Building,Corvallis, OR 97331, USA.
[3]Jet Propulsion Laboratory, California Institute of Technology, Pasadena, CA, 91109, USA

*Correspondence to:* Ranil Basnayake (rbasnaya@clarkson.edu)

**Abstract.** We illustrate the utility of variational destriping for ocean color images from both mulitspectral and hyperspectral sensors. In particular, we examine data from a filter spectrometer, the Visible Infrared Imaging Radiometer Suite (VIIRS) on the Suomi National Polar Partnership (NPP) orbiter, and an airborne grating spectrometer, the Jet Population Laboratory's (JPL) hyperspectral Portable Remote Imaging Spectrometer (PRISM) sensor. We solve the destriping problem using a variational

regularization method by giving weights spatially to preserve the other features of the image during the destriping process. The target functional penalizes 'the neighborhood of stripes' (strictly, directionally uniform features) while promoting data fidelity, and the functional is minimized by solving the Euler-Lagrange equations with an explicit finite difference scheme. We show the accuracy of our method from a benchmark data set which represents the Sea Surface Temperature off the Coast of Oregon, USA. Technical details, such as how to impose continuity across data gaps using inpainting, are also described.

**1  Introduction**

Striping is a persistent artifact in remote sensing images and is particularly pronounced in Visible-Near Infrared (VNIR) water-leaving radiance products such as those produced by operational sensors including NPP VIIRS, Landsat 8 Operational Land Imager (OLI), and Geostationary Ocean Color Imager (GOCI), as well airborne instruments such as NASA's JPL PRISM sensor. These sensors cover a temporal sampling range from daily (VIIRS) to hourly (GOCI), and spectral sampling from

multi-spectral (VIIRS, GOCI) to hyperspectral (PRISM). Striping is pronounced in products from all these sensors because atmospheric correction for ocean color products typically removes at least 90% of the signal recorded at the Top of Atmosphere (TOA). Put another way, any artifacts in the TOA signal are amplified by at least a factor of 10 in any derived water products such as normalized water leaving radiance of a specific spectral band (nLw($\lambda$)), or in product fields such as total suspended sediment (TSS) concentration maps.

Striping is ubiquitous and difficult to remove because it has many possible origins. The detectors themselves are subjected to small amplitude variations in both sensitivity and calibration. The view angles (azimuthal and zenith) also vary from detector to detector and pixel to pixel. Other differences in the instrument's optical path, components (e.g. mirrors), asynchronous readout, and so on, also cause striping. Not unexpectedly, the magnitude of the striping varies from image to image. Striping is





particularly problematic when comparing a sequence of images, since any difference computations between images produces spurious results in the neighborhood of stripes.

Ocean products from NPP VIIRS have shown problematic striping since its launch, which has led to focused efforts at both NASA and NOAA to find corrections. NASA created a vicarious destriping method for VIIRS images based on a collection of long term on-orbit image data, including solar and lunar calibrations. NASA's Ocean Biology Processing Group (OBPG) began serving operational products with their vicarious calibrations and destriping for VIIRS in 2014 Eplee et al. (2012). In contrast, a method for destriping VNIR images based on a single scene, using a hierarchical approach, was proposed in Bouali (2010). This particular variational method was more recently augmented with filtering using a hierarchical image decomposition Bouali and Ignatov (2013), and that algorithm has also been implemented by scientists at NOAA for operations with images from VIIRS Mikelsons et al. (2015, 2014). Scene based processing methods are advantageous for sensors which do not have dedicated calibration subsystems such as a solar diffuser, or where the data sets are limited in scope (such as airborne sensors) and do not include uniform scenes for vicarious calibration.

## 2   Regularization Destriping: the Functional and its Minimization

The method described here is closely related to the destriping functional described in Bouali (2010). Our work differs in its exact functional form, and its method of solution. In particular, we formulate a solution for destriping in an inverse-problem framework, and keep only the part of the functional in Bouali (2010) that smooths the stripes. This formulation allows us to provide an explicit numerical solution instead of an iterative one, the former being more suited for operational codes. We explicitly introduce a regularization parameter that controls the relative balance between the data term ("fitting the original image") and the regularity term ("smoothing out the stripes"). Solutions of this kind are common practice in inverse problems Vogel (2002), and fall under the rubric of Tikhonov regularization theory. As a further improvement to the destriping functional, we specifically define weights for the regularization term so that the algorithm applies only to the stripes while preserving the other features.

Assuming that the stripes are parallel to one another in the image plane, we take the direction of the stripes as the $x$ (horizontal) direction. Thus the data term representing the horizontal gradient difference between the original and the destriped images is given as

$$E_D(u) = \int_\Omega \left( \frac{\partial}{\partial x}(u - f) \right)^2 d\Omega, \tag{1}$$

where $\Omega$ is the image domain on $xy$ plane, $f(x,y)$ is the original image with stripes, and $u(x,y)$ is the destriped image.

The regularization term emphasizes the smoothness in the vertical direction, which is assumed to be free of stripes. This regularization term is given by

$$E_R(u) = \int_\Omega \left( \frac{\partial u}{\partial y} \right)^2 d\Omega. \tag{2}$$



The regularization parameter $\alpha > 0$ balances the data term and the regularization term. The resulting destriping functional is

$$E_C(u) = \int\limits_{\Omega} \left( \frac{\partial}{\partial x}(u-f) \right)^2 d\Omega + \alpha \int\limits_{\Omega} \left( \frac{\partial u}{\partial y} \right)^2 d\Omega. \tag{3}$$

This is the $x$-directional destriping functional proposed by Bouali in Bouali (2010), and it is equivalent to the basic form of our destriping functional when $\alpha = 1$. The choice of $\alpha$, as we show later, is key to achieving the balance of matching the original image and removing stripes. However, our approach differs from Bouali (2010) and our goal is to develop a destriping method which is easy to implement while preserving the other features of the image.

A drawback of scene-based detriping are unintended changes in the values of all the pixels and not just the stripes. If we apply the regularization term for the whole image as it is in Eq. (3), the entire image is effected. This could modify the original features of the image, in addition to recovering stripes. Therefore, we further develop our functional in Eq. (3) to regularize only the stripes. We introduce a mask $(L)$ to the regularization term, to limit the smoothing affects to the stripes. To obtain $L$ from the image, we first compute the slope of the image transverse to the stripes using first order finite differences. Then we sum the absolute differences parallel to the stripes. This yields the total value $(S)$ corresponding to each row. From the peaks of graph $S$ $vs$ $r$, where $r$ is the row index, we can identify the stripes and select a threshold to separate the stripes from the other features.

The mathematical expression for the computation of $S$ for an image $f$, of size $m$-by-$n$, with suitable a boundary condition, can be written as

$$S(r) = \sum_{c=1}^{n} |f(r,c) - f(r+1,c)|, \tag{4}$$

where $r = 1, 2, ..., m$ and $f(m+1, c)$ is the introduced boundary row. Now defining the threshold value from $S$, obtain the sparse matrix $L$ with ones indicating the locations of the stripes. Any row $r$, where $r = 1, 2, ..., m$ of matrix $L$ with size $m$-by-$n$ can be defined as

$$L(r,c) = \begin{cases} 1, & \text{if } threshold \geq S(r) \\ 0, & \text{otherwise,} \end{cases} \tag{5}$$

where $c = 1, 2, ..., n$.

Then the new destriping functional, with the spatially weighted regularization term, is written as

$$E(u) = \int\limits_{\Omega} \left( \frac{\partial}{\partial x}(u-f) \right)^2 d\Omega + \alpha \int\limits_{\Omega} L \left( \frac{\partial u}{\partial y} \right)^2 d\Omega. \tag{6}$$

The destriped image is obtained by minimizing the functional after choosing regularization parameter. Note that the functional $E(u)$ is invariant under constant shift. That is, $E(u+a) = E(u)$ for any constant $a$, implying that minimization of $E(u)$ leads to an infinite number of solutions. Because we want to keep the average intensity of the original and the destriped images the same, we assert $\langle u \rangle = \langle f \rangle$.





We create a destriped image by minimizing the energy functional in Eq. (6) using the Euler-Lagrange equation. For a functional of the form

$$J(u) = \int\limits_{\Omega} F(x, y, u, u_x, u_y) \, d\Omega,$$

on the bounded domain $\Omega$, the Euler-Lagrange equation is given as

$$\frac{\partial F}{\partial u} - \frac{\partial}{\partial x}\left(\frac{\partial F}{\partial u_x}\right) - \frac{\partial}{\partial y}\left(\frac{\partial F}{\partial u_y}\right) = 0. \tag{7}$$

Applying Eq. (7) to Eq. (6), the Euler-Lagrange equation, as explained in Vogel (2002); Basnayake and Bollt (2014) is the partial differential equation

$$u_{xx} + \alpha L u_{yy} = f_{xx}, \tag{8}$$

where subscripts represent the argument variable(s) of the partial derivatives. We can rewrite the Eq. (8) as

$$(D_{xx} + \alpha L D_{yy}) u = D_{xx} f, \tag{9}$$

where, the operators $D_{\bullet\bullet}$ are two dimensional arrays of size $k \times k$ used to compute the partial derivatives of a given vector of size $k \times 1$ with respect to the indices $\bullet\bullet$.

We use finite difference approximations with suitable boundary conditions for each derivative to directly represent these differential operators. In Eq. (9) we stack the given image of size $p \times q$ onto $k \times 1$ vector, where $k = pq$. We can now use the differential operators to write the linear Euler-Lagrange equation in the form of $Au = b$, and solve for $u$ using an appropriate numerical method rather than solving the Euler-Lagrange equation for $u$ from an iterative scheme such as Gradient decent method.

## 2.1 Construction of the Differential Operator

We construct the operator $D_{xx}$ using finite difference approximations. The operator $D_{yy}$ is built by taking the transpose of the finite difference stencil. Suppose we have a function $M(x, y) \in \mathbb{R}^{p \times q}$. We need to compute the second order partial derivative of $M$ with respect to $x$. We use a fourth-order finite difference approximation and compute the point-wise second partial derivative of the array $M$ with respect to $x$.

As an example, take an array $M(x, y)$ of size $3 \times 5$ where we want to compute $M_{xx}(x, y)$. We index the elements in the array in the form of a column vector as shown in Table 1, with two added boundary columns for each side.

The boundary points are highlighted in red. If we compute the partial derivative of $m_1$ with respect to $x$, the resulting approximation is obtained as

$$\frac{\partial^2 m_1}{\partial x^2} = \frac{1}{12h^2}\left[-\mathbf{m_4} + 16\mathbf{m_1} - 30m_1 + 16m_4 - m_7\right] = \frac{1}{12h^2}\left[-14m_1 + 15m_4 - m_7\right].$$





**Table 1.** An array of $3 \times 5$ with boundary points in red

| $\mathbf{m_4}$ | $\mathbf{m_1}$ | $m_1$ | $m_4$ | $m_7$ | $m_{10}$ | $m_{13}$ | $\mathbf{m_{13}}$ | $\mathbf{m_{10}}$ |
|---|---|---|---|---|---|---|---|---|
| $\mathbf{m_5}$ | $\mathbf{m_2}$ | $m_2$ | $m_5$ | $m_8$ | $m_{11}$ | $m_{14}$ | $\mathbf{m_{14}}$ | $\mathbf{m_{11}}$ |
| $\mathbf{m_6}$ | $\mathbf{m_3}$ | $m_3$ | $m_6$ | $m_9$ | $m_{12}$ | $m_{15}$ | $\mathbf{m_{15}}$ | $\mathbf{m_{12}}$ |

**Table 2.** A discretized derivative operator $D_{xx} \times 12h^2$ for a $3 \times 5$ matrix.

| | $m_1$ | $m_2$ | $m_3$ | $m_4$ | $m_5$ | $m_6$ | $m_7$ | $m_8$ | $m_9$ | $m_{10}$ | $m_{11}$ | $m_{12}$ | $m_{13}$ | $m_{14}$ | $m_{15}$ |
|---|---|---|---|---|---|---|---|---|---|---|---|---|---|---|---|
| $m_1$ | -14 | 0 | 0 | 15 | 0 | 0 | -1 | 0 | 0 | 0 | 0 | 0 | 0 | 0 | 0 |
| $m_2$ | 0 | -14 | 0 | 0 | 15 | 0 | 0 | -1 | 0 | 0 | 0 | 0 | 0 | 0 | 0 |
| $m_3$ | 0 | 0 | -14 | 0 | 0 | 15 | 0 | 0 | -1 | 0 | 0 | 0 | 0 | 0 | 0 |
| $m_4$ | 15 | 0 | 0 | -30 | 0 | 0 | 16 | 0 | 0 | -1 | 0 | 0 | 0 | 0 | 0 |
| $m_5$ | 0 | 15 | 0 | 0 | -30 | 0 | 0 | 16 | 0 | 0 | -1 | 0 | 0 | 0 | 0 |
| $m_6$ | 0 | 0 | 15 | 0 | 0 | -30 | 0 | 0 | 16 | 0 | 0 | -1 | 0 | 0 | 0 |
| $m_7$ | -1 | 0 | 0 | 16 | 0 | 0 | -30 | 0 | 0 | 16 | 0 | 0 | -1 | 0 | 0 |
| $m_8$ | 0 | -1 | 0 | 0 | 16 | 0 | 0 | -30 | 0 | 0 | 16 | 0 | 0 | -1 | 0 |
| $m_9$ | 0 | 0 | -1 | 0 | 0 | 16 | 0 | 0 | -30 | 0 | 0 | 16 | 0 | 0 | -1 |
| $m_{10}$ | 0 | 0 | 0 | -1 | 0 | 0 | 16 | 0 | 0 | -30 | 0 | 0 | 15 | 0 | 0 |
| $m_{11}$ | 0 | 0 | 0 | 0 | -1 | 0 | 0 | 16 | 0 | 0 | -30 | 0 | 0 | 15 | 0 |
| $m_{12}$ | 0 | 0 | 0 | 0 | 0 | -1 | 0 | 0 | 16 | 0 | 0 | -30 | 0 | 0 | 15 |
| $m_{13}$ | 0 | 0 | 0 | 0 | 0 | 0 | -1 | 0 | 0 | 15 | 0 | 0 | -14 | 0 | 0 |
| $m_{14}$ | 0 | 0 | 0 | 0 | 0 | 0 | 0 | -1 | 0 | 0 | 15 | 0 | 0 | -14 | 0 |
| $m_{15}$ | 0 | 0 | 0 | 0 | 0 | 0 | 0 | 0 | -1 | 0 | 0 | 15 | 0 | 0 | -14 |

Continuing this manner for all the elements, we can compute the differential operator $D_{xx}$. Using a multiplication factor of $12h^2$, we obtain a sparse matrix with only five non-zero diagonals (Table 2).

Similarly, $D_{yy}$ and the other derivatives can be estimated as needed. The boundary points are highlighted as bold entries. If we compute the partial derivative of $m_1$ with respect to $x$, the resulting approximation is obtained as

$$5 \quad \frac{\partial^2 m_1}{\partial x^2} = \frac{1}{12h^2}\left[-\mathbf{m_4} + 16\mathbf{m_1} - 30m_1 + 16m_4 - m_7\right] = \frac{1}{12h^2}\left[-14m_1 + 15m_4 - m_7\right].$$

Continuing this manner for all the elements, we can compute the differential operator $D_{xx}$. Using a multiplication factor of $12h^2$, we obtain a sparse matrix with only five non-zero diagonals (Table 2).



## 2.2 Solution to the Euler-Lagrange Equation

Now we can determine the solution to Eq. (9). We rewrite the Eq. (9) as

$$Au = b,\tag{10}$$

where $A = D_{xx} + \alpha L D_{yy}$ and $b = D_{xx}f$. If the size of the given striped image $f$ is $p \times q$, then $A$ is a $k \times k$ sparse array and $b$

is a $k \times 1$ array, where $k = pq$.

Using a suitable value for the regularization parameter, Eq. (10) can be solved as a linear system. The system is sparse and hence the computation time for an image with $n$ pixels is of $O(n)$ for each iteration. Clearly, at this stage, for a given $\alpha$, Eq. (10) is straightforward to solve, however in terms of the image processing, the specific choice of $\alpha$ plays an important role. To achieve "the most appropriate solution," we need to determine the best regularization parameter $\alpha$.

## 2.3 Selection of the Regularization Parameter

The condition number of the resulting matrix quantifies the amplification of computational errors seen while solving the problem by direct computation. The condition number may be computed as the ratio between the largest singular value and the smallest singular value of the coefficient matrix. If the condition number is large, then the coefficient matrix is said to be ill-conditioned and hence the corresponding system is ill-posed. In an ill-posed system, the solution is highly sensitive to

perturbations of the input data. Regularizing an ill-posed system, emphasizing a desired property of the problem, introduces a stable way to define a desirable solution Vogel (2002); Hansen et al. (2006); Hansen (2010). This is the standard trade-off between regularity and stability in Tikhonov regularization terms.

We regularize our computed solution by emphasizing the expected physics. To damp the accumulated errors from the residuals, we must make sure that we add sufficient regularity. The balance between the data term and the regularization term is

very important: if we add too much regularity, it will divert the solution from the desired solution. Stated in terms of Tikhonov regularization, $\alpha$ serves the role to select a unique optimizer $u$, from what would be and otherwise ill-posed system had only the data fidelity had been chosen. In terms of the images, the data fidelity states that the optimizer image $u$ should "appear as" the original data image, measured here in terms of along the stripes, but "smoothed" according to the regularizer term, in this case transverse to the stripes. The question then is how to balance these two effects.

Some of the common methods to determine the regularization parameter in inverse problems are the $L$-curve method Hansen (1992, 2000) and $U$-curve method Krawczyk-StańDo and Rudnicki (2007); Krawczyk-Stado and Rudnicki (2008). After trials with both methods we selected the $U$-curve method. Starting with the functional

$$J(u) = \arg\min_{u} \|Au - z\|_2^2 + \alpha \|u\|_2^2,\tag{11}$$

a regularized solution is selected for a fixed $\alpha$, and the norm of the residuals and solution is written as

$$
\begin{aligned}
x(\alpha) &= \|Au_\alpha - z\|_2^2 \text{ and} \\
y(\alpha) &= \|u_\alpha\|_2^2.
\end{aligned}\tag{12}
$$





Considering a range of $\alpha$ values, and computing the sum of the reciprocals, results in a 'U' curve,

$$U(\alpha) = \frac{1}{x(\alpha)} + \frac{1}{y(\alpha)}. \tag{13}$$

The 'best' $\alpha$ value corresponds to the minimum of $U(\alpha)$. The best $\alpha$ value must be in the interval $\alpha \in \left(\sigma_1^{2/3}, \sigma_r^{2/3}\right)$, where $\sigma_1$ is the largest singular value of the operator $A$, and $\sigma_r$ is the smallest non-zero singular value of the operator $A$ Krawczyk-StańDo and Rudnicki (2007).

### 2.4   Inpainting Data Gaps

Unlike terrestrial images, which can show sharp edges, ocean color images typically appear continuous. This is because the water tends to diffuse any color agents in the water column, and the spatial resolution of sensor is usually finer than those diffusive features. The same holds spectrally if the sampling wavelength is less than the autocorrelation function of the spectra, as is the case for hyperspectral images. However, this continuity is broken if gaps appear in the data.

Clouds are very bright and often saturate the sensor. In normal processing clouds are typically masked from the data since their large radiance values obscure the (relatively dark) ocean. These type of processing masks also cause large, irregular, data gaps. There are other sources of data gaps as well, as discuss here, that can be inherent in the sensor design.

As an example of data gaps introduced by system design, consider VIIRS data, which uses 16 detector elements to generate each spatial image. The spatial footprint of each adjacent sensor element overlaps at the detector edgesCao et al. (2013). To reduce data transmission from the satellite to ground stations, the overlapping regions are not transmitted, causing the so-called 'bow-tie' effect. This appears as visible horizontal stripes in the Level 1 or Level 2 unmapped pixel values. These gaps are removed during projection to ground based coordinates, so-called Level-3 data (L3). However, stripes which are linear in the 'unmapped' data become nonlinear after mapping, and are more difficult to remove. Therefore we work with the Level 1 and Level 2 data where the stripes are linear, and often aligned with the focal plane array or detector elements.

We need to preprocess the image data in such a way as to insure continuity across any data gaps. An obvious way to fix the gaps is to use 'inpainting,' a technique from image processing – not to infer what the actual missing data might be, rather simply to impose continuity across the whole image when gaps are present. In a museum setting, inpainting refers to the process whereby a painter-restorer interprets a damaged painting by artistically filling in damaged or missing parts of a painting, smoothly bleeding in the colors of the painting that surrounds the damage, Bertalmio et al. (2011). In modern day computational parlance, inpainting refers to an image processing algorithm that mimics this idea. A wide variety of methods have been implemented Bertalmio et al. (2000).

We do nothing fancy here with inpainting. Missing data is filled in by solving the Laplace equation with Dirichlet boundary conditions, $\nabla u = 0$ in $\Omega$ and $u = f$ on $\partial\Omega$, specially we use the *Matlab* routine *roifill*. Because the algorithm approximates the partial derivatives using finite difference approximations, inpainting should be done before applying the destriping algorithm on images.

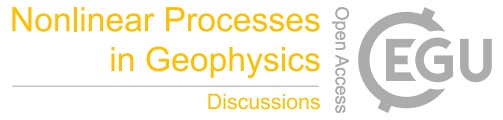

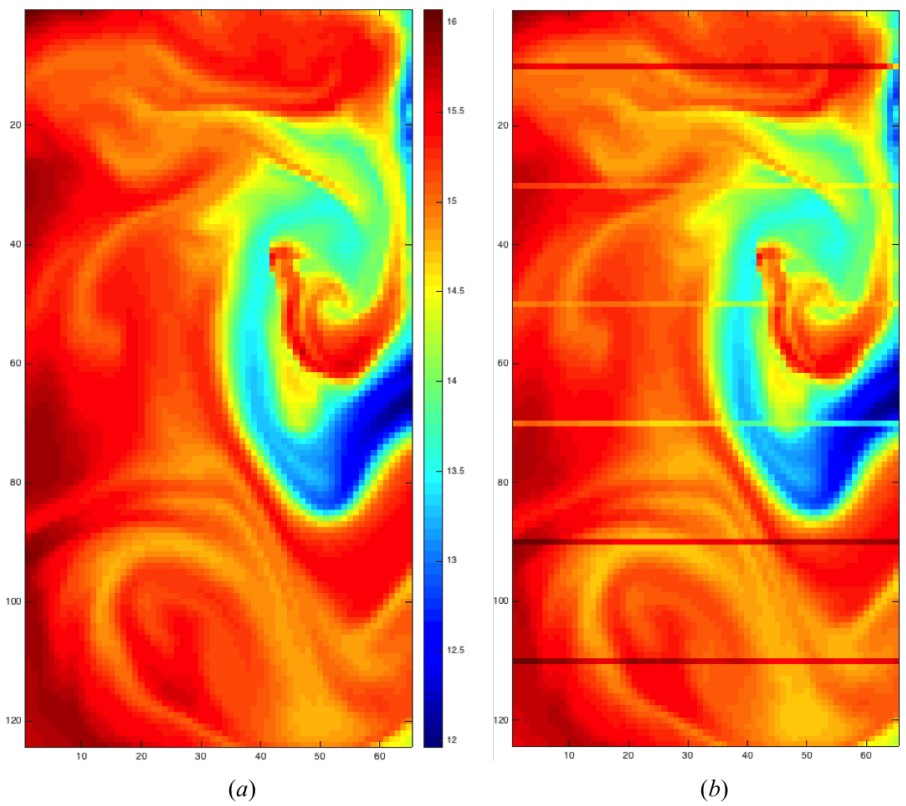

**Figure 1.** Image (a) shows (simulated) variations of Sea Surface Temperature off the coast of Oregon, USA on 1 August 2002. The image was generated from a Regional Ocean Model System (ROMS, Courtesy of John Osborne, Oregon State University), using the data assimulation from the Geostationary Operational Environmental Satellite (GOES). Image (b) is created by adding artificial stripes every 20th row.

## 3 Results and Discussion

In this section, we first apply the destriping method to a simulated image of Sea Surface Temperature (SST), and then apply the destriping algorithm on two real images, one from the multi-spectral NOAA imager VIIRS, and the second from the JPL PRISM hyperspectral sensor.

## 3.1 Benchmark data: Sea Surface Temperature

A benchmark data set – Sea Surface temperature data off the Oregon coast – was used to test the codes. The original image data is shown in Fig. 1(a), where red represents high temperature and blue represents the low temperature. This simulated image is from a $3 - D$ Regional Ocean Model System (ROMS) showing the Oregon Coast on 1 August 2002 Osborne et al. (2011). This area covered is from North Latitude 41 to 46 degrees and West longitude from -124 to -125.





The image (b) in Fig. 1 shows the artificially added stripes on the image shown in Fig. 1(a). The intensities of stripes are assigned as the absolute vales of difference between the original intensity and the average of the each striped row. However, stripes at $10^{th}$, $90^{th}$ and $110^{th}$ rows have some added noise intensity values to visualize them properly.

Our next step is to apply the destriping method to the SST data and check the accuracy of the algorithm. There, we compare the solutions with regularization parameters $\alpha = 1$ and $\alpha = 3 \times e^{-1}$ using the functional with un-weighted regularization term as shown in Eq. (3) and $\alpha = 7 \times e^{-3}$ with spatially weighted regularization term as shown in Eq. (6). The idea of this comparison is to show the importance of the selection of our regularization method and how the weighted regularization term improves the destriping results. The destriped image from the un-weighted regularization term is shown in Fig. 2 (a) and

clearly it is over smooth as most of the original features are destroyed from the destriping. Therefore, we chose an appropriate $\alpha$ value to balance data term and the un-weighted regularization term from the $U$-curve method. The $U$-curve solution is $\alpha = 3 \times e^{-1}$ and the resulting destriped image is shown in Fig. 2 (b). However, it can be observed that some features of the destriped image have undergone smoothing at some places in addition to the stripes during the destriping process. Hence it is an important task to preserve the other features where there are no stripes while applying destriping. We achieve this by

improving the regularization term by incorporating different weights to the places where stripes are available and not. The weighted matrix $L$ can be obtained from the expression as we explained in Eq. (5) giving zeros to the places where no stripes. In this manner, we can preserve the features of the original image from smoothing and the resulting image is shown in Fig. 2 (c).

Next task is to check the accuracy of the estimated values for the stripes. The stripe at $110^{th}$ row was randomly selected for

this purpose. There we plot the intensity values of the stripe (black), reconstruction (red) and the actual values as they were against column index. The results from the functional in Eq. (3) with $\alpha = 1$ is shown in the graph (c) of Fig. 3. The reconstructions of the first 20 pixels and the last 10 pixels are closer to the original values, but the rest has significant differences compared to the actual value. The graph (b) in Fig. 3 shows destriped results of the $110^{th}$ row from the functional in Eq. (3) using the $U$-curve solution, $\alpha = 3 \times e^{-1}$. In this case, the reconstruction of some middle pixels ($18 - 28$ and $31 - 37$) and the

last 10 pixels are off by some extend, but the rest is much closer to the actual values than the construction with $\alpha = 1$. The graph (c) in Fig. 3 shows the best solution among these approachers and it is obtained from the weighted regularization term with $\alpha = 7 \times e^{-1}$. This reconstruction very closer to the actual values and hence it is the most accurate reconstruction. This is the sort of results exactly we expect from a destriping algorithm. More importantly, if we compare 'a stripe like feature' at the $18^{th}$ row, we can conclude that the importance of giving weights to the regularization term as the weighted regularization term

focus only the stripes.

In addition to the reconstruction of stripes, we need to pay attention to the rest of the features of the image. The idea of destriping is to estimate the values for stripes and preserve the other original features of the image. Therefore, we randomly select the $67^{th}$ row of the image to compare before and after effects of destriping at a place where there is no actual stripe. The

graphs (a) and (b) in Fig. 4 are from the functional in Eq. (3) with $\alpha = 1$ and $\alpha = 3 \times e^{-1}$, respectively and the graph (c) in





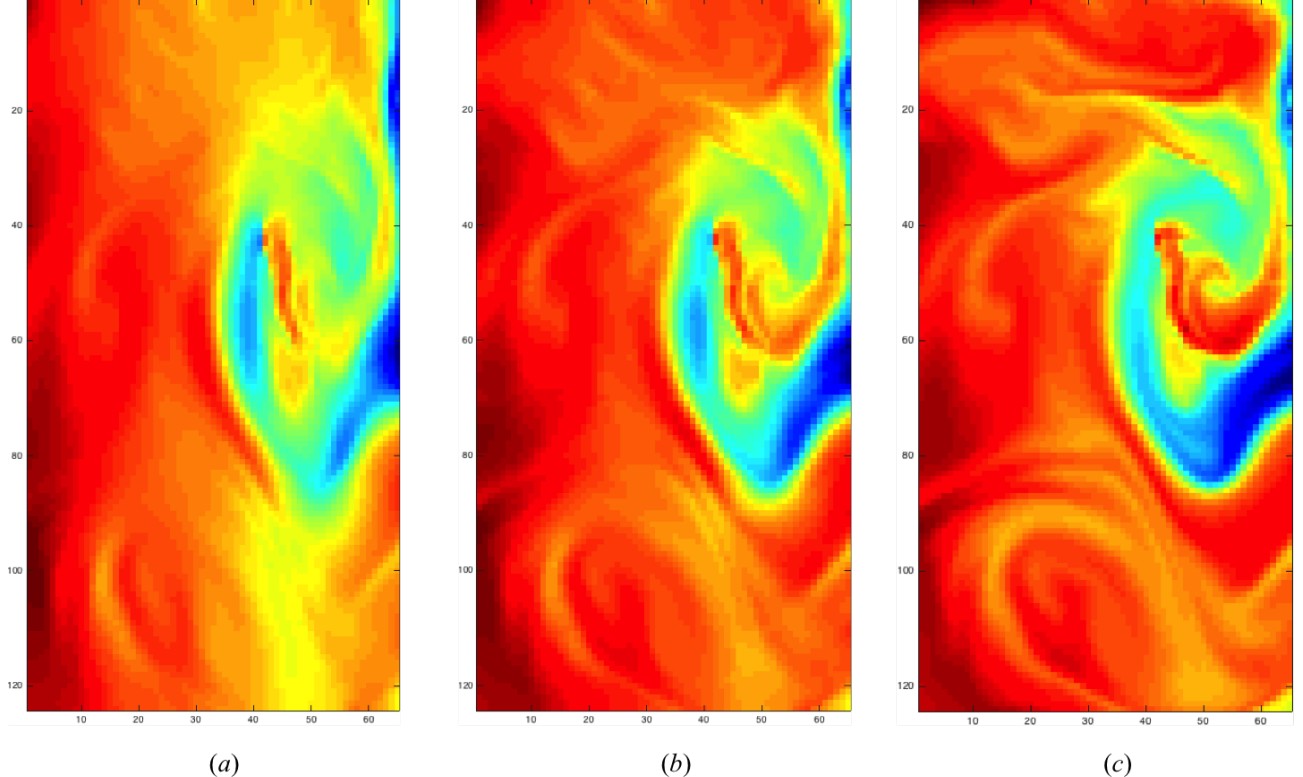

$$(a) \qquad\qquad\qquad (b) \qquad\qquad\qquad (c)$$

**Figure 2.** These three images represent the destriped image of the image shown in Fig. 1(b) from three different ways. Image (a) and (b) are obtained with $\alpha = 1$ and $\alpha = 3 \times e^{-1}$ from the functional in Eq. (3). Image (c) is obtained with $\alpha = 7 \times e^{-1}$ from the functional with spatially weighted regularization term as shown in Eq. (6).

Fig. 4 from the functional in Eq. (6) with $\alpha = 7 \times e^{-1}$. The graphs (a) and (b) in Fig. 4 conclude that the destriping has affected the other features of the image when the functional in Eq. (3) is applied regardless of the value of the regularization parameter. However, the graph (c) in Fig. 4 shows that the functional with the spatially weighted regularization term in Eq. (6) does not affect the other features of the image.

The $S$ curve corresponding to the SST image is shown in graph (a) in Fig. 5. The "threshold" value for this problem is also shown on same graph and it is 12 in this case. The peak points that has crossed by the threshold value are the stripes. As a comparison of the full destriping image, the absolute percentage error between the striped and destrieped image is shown in the image (b) in Fig. 5. This is computed by,

10  Absolute Percentage Error $= \dfrac{|f - u|}{f} \times 100,$

where $f$ and $u$ are striped and destriped images respectively. The graph of the 'absolute percentage error shows that not only $67^{th}$ row, but also all the other non-stripe rows are not affected from this spatially weighted regularization method.

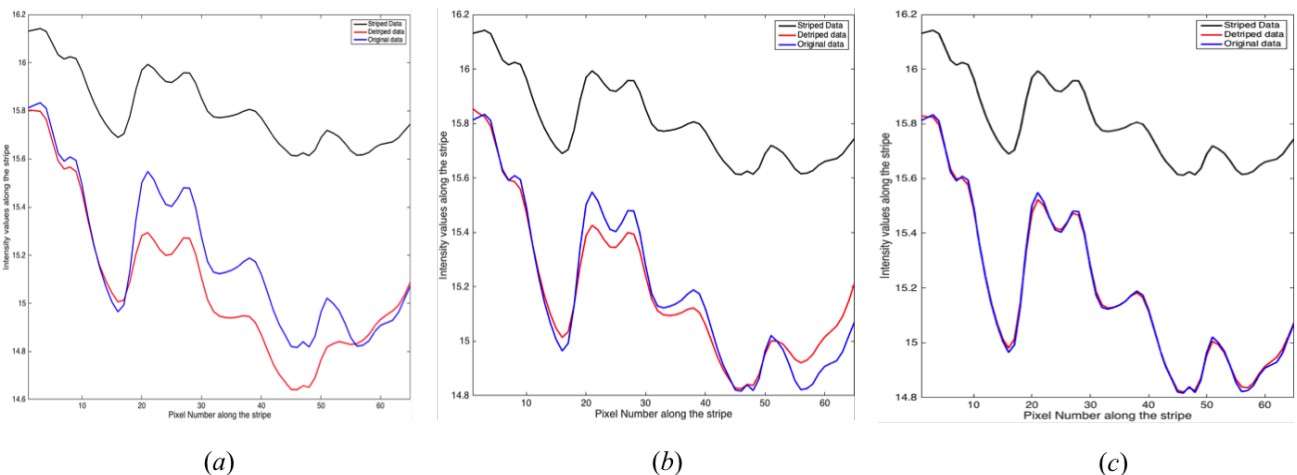

$(a)$          $(b)$          $(c)$

**Figure 3.** This figure presents three different reconstructions of the stripe at the $110^{th}$ row of the image shown in Fig. 1(b). The graphs show the actual data in blue, striped data in black and the destriped data in red. Graphs (a) and (b) represent the reconstructions from the functional in Eq. (3) with $\alpha = 1$ and $\alpha = 3 \times e^{-1}$, respectively. The graph (c) shows the reconstruction from the functional in Eq. (6) with $\alpha = 7 \times e^{-1}$.

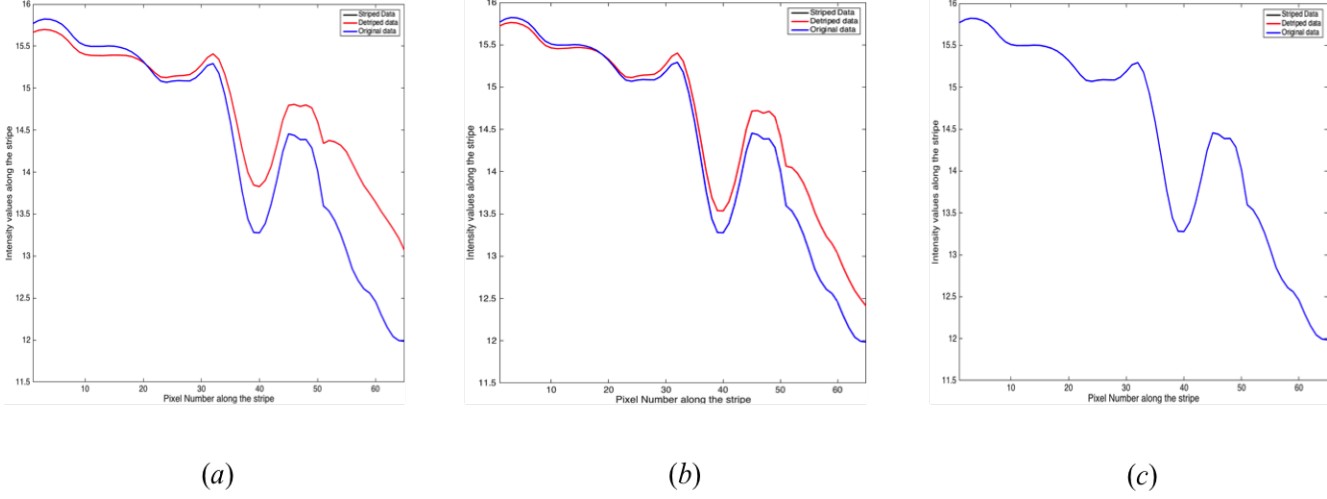

$(a)$          $(b)$          $(c)$

**Figure 4.** This figure shows the effects of destriping on the places where 'no stripes'. We randomly picked the $67^{th}$ row for this comparison. When $\alpha = 1$ in the un-weighted regularization functional, the image is more smoothed and affects destriping to the whole image. Much better results can be obtained from the the un-weighted regularization functional with $\alpha = 3 \times e^{-1}$ which is the $U-$curve solution and shown in graph (b). Spatially weighted regularization term with $\alpha = 7 \times e^{-1}$ provides less effect to the other features of the image and we can observe that from the graph (c).



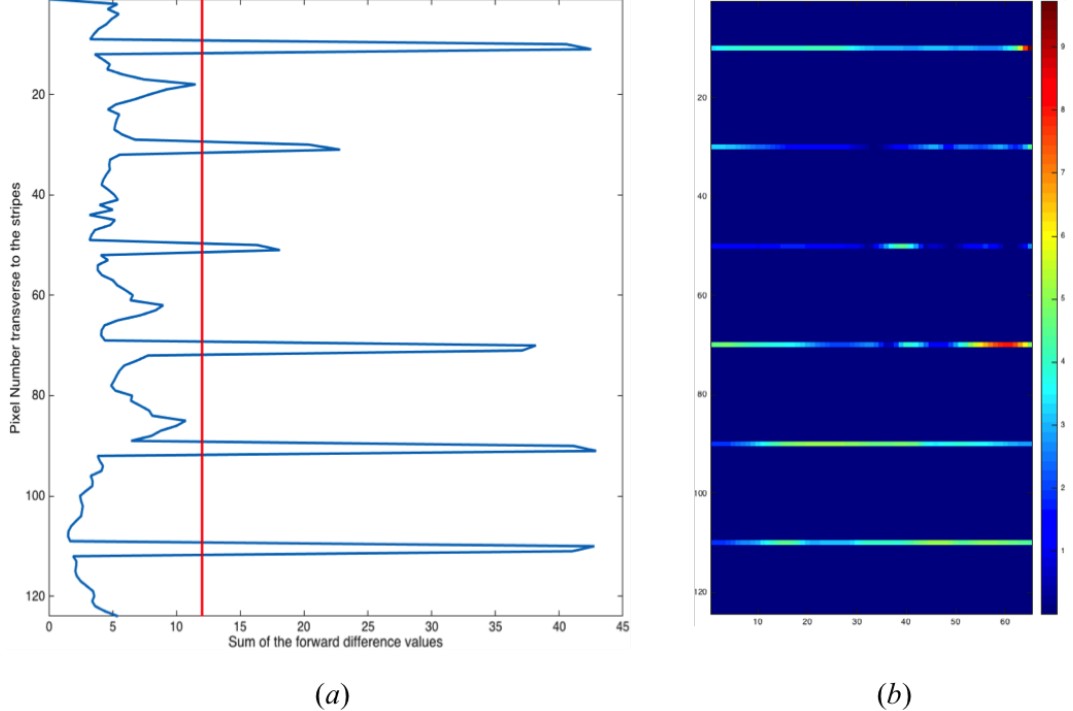

$$(a) \qquad\qquad (b)$$

**Figure 5.** Image (a) shows the $S$ function values against the column numbers. The peak points represent the stripes. When "treshold" is set at 12, only stripes can be included for the regularization but excludes all the other features of the image. Then the percentage error between the striped and destriped image are shown in the image (b). The error where there are no stripes is always closer to the zero

## 3.2 Example 2 - VIIRS images

A good example of VIIRS stripping is shown ( Fig. 6) in a chlorophyll concentration map near the Santa Monica region in Southern California on November 07, 2014 NAS. The chlorophyll concentration is given in $mg/(m^3)$. Green is the land and the dark blue is the dropped data and the missing data. The full VIIRS data granule covers $-122.09°$ W to $-116.90°$ E, and 34.2° N to 31.6° S ( Fig. 6). We consider a subsetted (cropped) region of interest to highlight the stripes in Fig. 7 (a).

The first step of applying this destriping method is the determination of the threshold value to separate the neighborhood of stripes and rest of the features. However, when we deal with real data, we may not always get nice and smooth images. For instance, if we compute the $S$ curve values using the whole image, we are unable to get any evidence to determine the threshold value as sum of the magnitude of forward differences in some other rows are higher than the that of stripes. Therefore, we need to carefully pick a subregion of the image that includes all the rows with some selected columns. The sum of the magnitude of forward differences in non-stripe rows should be less than the that of stripes in this region. Then the stripes can be easily





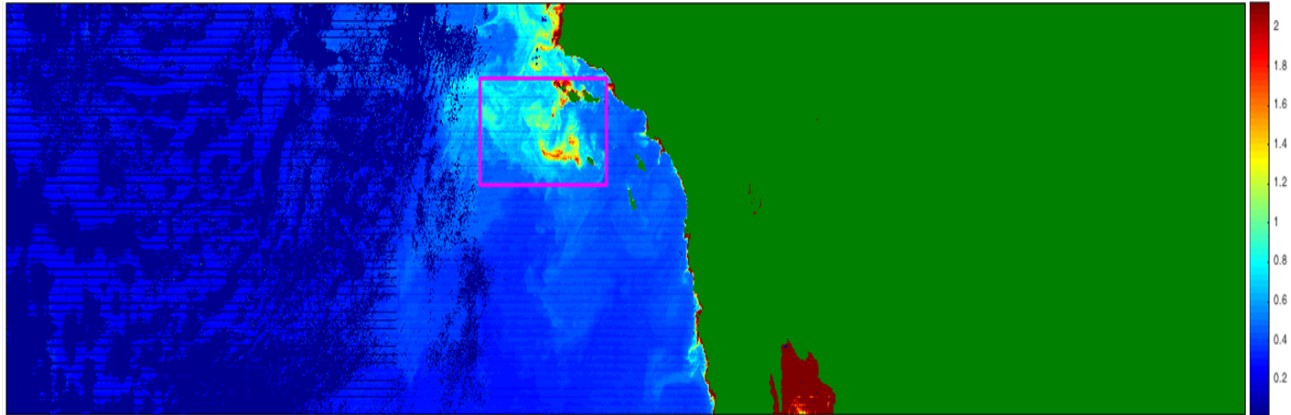

**Figure 6.** The image shows the chlorophyll concentration in $mg/(m^3)$ near the Santa Monica region in Southern California as viewed by VIIRS on November 07, 2014. Green represents the land and dark blue represents the dropped data due to bow-tie effects and missing data due to clouds. For a detailed discussion, we next consider the subset of the image that is covered by the pink square in the image.

highlighted. For this example, we select the region using the columns from 137 to 148 and it is shown in Fig. 7 (b) with the $S$ curve. The threshold value can be determined from the $S$ curve and it is $0.558$ for this image.

The effects of spatially weighted regularizing destriping are shown in Fig. 7 (e). We apply our algorithm to the image shown in Fig. 7 (a). In this case, the regularization parameter was $\alpha = 10^{-2}$ with the weighted regularization term. The intensity of destriped image now varies smoothly. Therefore, the image is smooth enough to further post process for other applications such as computing optical flow. The approach proposed in Bouali (2010) without the transverse direction functional, effectively sets $\alpha = 1$ in our functional Eq. (3). However, our destriping functional as shown in Eq. (6) has weight-matrix $L$ inside the regularization term and hence the two approaches are not the same. Image (c) in Fig. 7 represents destriped image when $\alpha = 1$ without the weighted regularization term, where image (a) is the original image. The destriped image is blurred and we lose some information from the original image due to over regularization. In this case, by "over regularization", we mean that continuity in the transverse direction to the "stripes" has been emphasize so strongly in functional Eq. (3) when $\alpha$ is close to 1, that it is past balance against the need to also emphasize the image data in the first term called "data fidelity". On the other hand, image (d) is the destriped image of image (a) with $\alpha = 10^{-5}$ in weighted regularization term. In this image, we still can observe some stripes due to less regularity to smooth the stripes. Hence, it is clearly observed that a given weighting factor to the regularization term is necessary and also choosing the best regularization parameter is very important. We can observe this scenario when we compare the images (c), (d) and (e) in Fig. 7 as $\alpha$ varies from $10^{-5}$ to 1. Therefore, once we have an appropriate $\alpha$, we do not need an extra functional to reduce the blurring effects as explained in Bouali (2010) and hence our approach provides a simple algorithm to remove the stripes. The appropriate $\alpha$ is selected from the $U$-curve method as explain in Sec. 2.3. Image (f) in Fig. 7 shows the percentage error between the striped and destriped images. This a proof to conclude that the effects from the destriping on the image where there are no stripes is negligible.

**Figure 7.** Image (a) shows the cropped region shown in Fig. 6 and the graph (b) shows the $S$ curve with the threshold and the image piece that we used compute the values of $S$ curve. The images (c), (d and e) represent the destriped images of (a) with $\alpha = 1$ with unweighted regularization term and $\alpha = 10^{-5}$ and $\alpha = 10^{-2}$ with weighted regularization term respectively. Image (e) provides the best solution for the destrped image. Image (c) is over regularized whereas image (d) is not sufficiently regularized. Image (f) represents the percentage error between images (a) and (e).

Image based methods such as the variational destriping can be used together with other destriping methods. For instance, NASA's vicarious calibration of L2 (*.nc) products method uses a monthly moon calibration to monitor the striping and



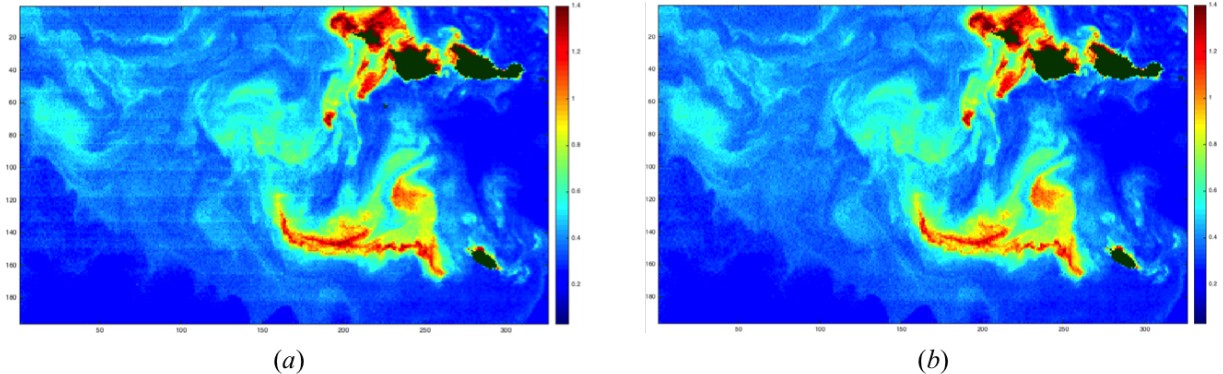

$(a)$                                         $(b)$

**Figure 8.** Image (a) shows the destriped image scene of the image (a) in Fig. 7 from NASA's vicarious calibration of L2 (*.nc) products method. While the NASA's vicarious calibration of L2 (*.nc) products method does improve over the raw image, there are still stripe artifacts present. The following image (b) represents the destriped image of the image (a) from our method.

calibrations, and create up to date corrections using the entire image collection. Fig. 8 (a) shows the VIIRS image in Fig. 7 (a) after the NASA correction and the data is publicly available at NAS. However, the striping artifacts are still visible in that image. Fig. 8 (b) shows the application of the weighted regularizing destriping to the image from the NASA's vicarious calibration of L2 (*.nc) products method. The product from the applications of both corrections is superior to either individual

correction. The The regularization parameter for this image was $5 \times 10^{-3}$ and the destriped image is shown in Fig. 8 (b). The threshold value for the columns 137 to 148 was $0.5$.

When we compare the images (a) and (b) in Fig. 8, it can be observed that our method can be used to further improve upon the destriped images, after the NASA's vicarious calibration of L2 (*.nc) products method has already been applied.

### 3.3   Example 3 - JPL PRISM Images

In the last example, we apply the variational destriping algorithm to another data set from the JPL PRISM hyperspectral imager, where the data is publicly available at JPL. Stripes arise in this sensor because of the focal plan array read out mechanism has cross-talk artifacts. An image from the band centered at 410 nm (band 22) is shown in Fig. 9 which was taken from an airborne campaign around Monterey, CA near the Eklhorn Slough Pantazis et al. (2015).

The regularization parameter to destripe the image (a) in Fig. 9 was $\alpha = 5 \times 10^{-3}$. The destriped image is shown in image

(b) in Fig. 9. We used the columns from 50 to 70 to determine the threshold value to assign the weights in the regularization term and it was $1.7$ for this image. It can be clearly observe that the stripes are smoothed from the destriping method while preserving the other features of the image.





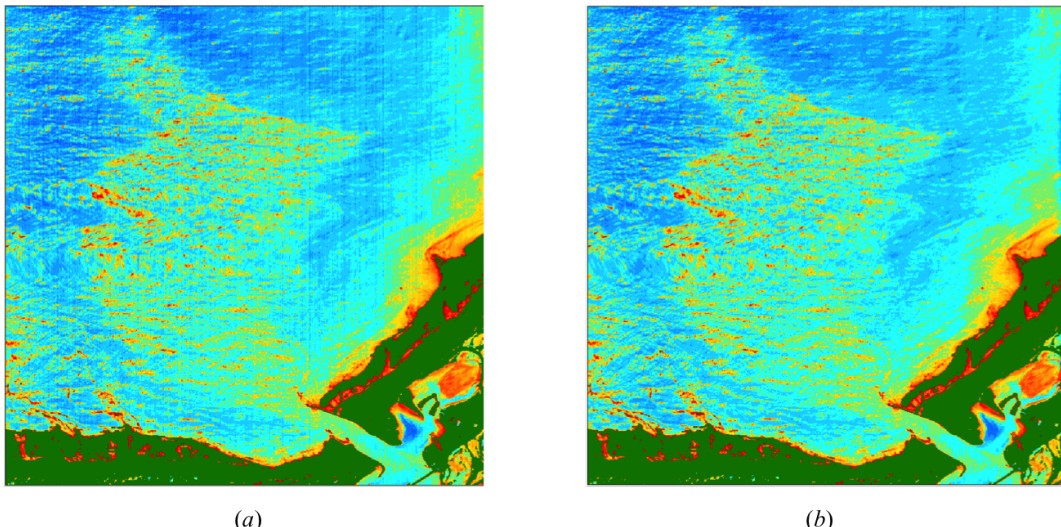

(a)                                           (b)

**Figure 9.** Image (a) is band 22 ($\tilde{4}$10 nm) of a hyperpectral image which was taken from JPL PRISM JPL. The stripe pattern is vertical and the destriped images shown in image (b). Green represents the land of the observed region.

## 4   Conclusions

We present a variational destriping method by explicitly including a tunable regularization parameter with a weighted regularization term to a part of the destriping functional in Bouali (2010). In other words, we modeled one piece of destriping functional in Bouali (2010) into a standard variational based approach employing the Tikhonov regularization theory Vogel

5   (2002). According to the Tikhonov regularization theory, the tuning parameter allows us to properly balance the effects of optimizing the data fidelity and the smoothing effects of regularization term with the help of given weights. The introduction of the weighted regularization term avoids the effects on the original features of the image during the process of destriping. As a preprocessing step, we apply an image processing technique to avoid the error accumulation from the "bow-tie" effects and missing data due to clouds. We also demonstrate alternative numerical aspects to implementing these methods which allows

10   us to write the solver in terms of common matrix manipulations. Lastly, we show that applying a scene based method to the destriped VIIRS L2 product from NASA calibration method results in additional improvements in scene uniformity.

*Author contributions.* Ranil Basnayake, Erik Bollt and Jie Sun developed the variational based destriping algorithm by introducing a weighted regularization term to the traditional destriping functional. Nicholas Tufillaro collected raw data from reliable sources and processed the data readable in Matlab. Michelle Gierach provided the JPL-PRISM data.

15   *Competing interests.* The authors of this work have no competing interests.



*Acknowledgements.* The authors Ranil Basnayake, Erik Bollt, Nicholas Tufillaro and Jie Sun were supported by the National Geospatial Intelligence Agency under grant number HM02101310010. Also Erik Bollt was supported by, the Office of Naval Research: PI, N00014-15-1-2093.





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
