# Peer review of "Regularization Destriping of Remote Sensing Imagery"

_Nonlinear Processes in Geophysics, 2016_

## Referee Comment (RC1) · K. McIlhany (Referee) · 9 Feb 2017

I found the paper interesting and relevant. There are numerous grammatical errors however which need to be addressed. There are enough that I simply scanned my edited copy for your perusal. My strong suggestion is that one of the writers read aloud the paper so that these mistakes are caught early on before submitting in the future. I realize that some of the authors are non-native English speakers and I am always impressed at how well they perform in the scientific environment, however, technical writing requires clarity. In some cases, the grammar issues lead to confusion about what exactly is being said.

On pages 4-5 there is come confusion in the text about whether the equations indicated boundary points as either "red" or "bold". Further, this confusion is present in table 1.

The PDF file does not show "red" when it is indicated in table 1 but does show red in table 2. When printing the article in black and white, it is better to indicate these as bold, so perhaps they could be both red and bold and then everyone is happy. Also, on page 4 and 5 is repeated text, pick one and remove the other.

Overall, I enjoyed the paper and its content. I do have a few technical questions, but nothing to prevent publication.

1) Page 7: In seeking a minimum for the U-curve method, how many numerical attempts at alpha are reasonable? 2) Page 9: When the data contains a horizontal stripe, do you accidentally remove it (like a road) and if not, how do you know to avoid it? 3) Page 10: Why 12 for the threshold? How do you choose 12? A suggestion would be to turn Fig. 5 by 90 degs and then set the threshold to the FWHM of the highest peak found. You could repeat this process using the next lowest peak and see how many more stripes are found. Eventually, you will see a rise in the number of stripes found and then raise the threshold to an earlier value when the number of stripes found was more consistent. 4) When citing "inpainting" did you consider Andrea Bertozzi?

Most of my grammatical issues are highlighted in red ink on the attached PDF. I answered my own inquiries about the "argmin_u" and "vicarious" comments, so please disregard those, but the rest are genuine issues about writing.

Thank you for the opportunity to review your work,

Kevin Mcilhany

Please also note the supplement to this comment:
http://www.nonlin-processes-geophys-discuss.net/npg-2016-74/npg-2016-74-RC1-supplement.pdf

---

## Author Comment (AC1) · 23 Mar 2017

article a4wide color graphicx float float plaintop table

**Responses to the Comments made by Prof. K. McIlhany**

Ranil Basnayake, Erik Bollt, Nicholas Tufillaro, Jie Sun, and Michelle Gierach

March 23, 2017

Dear Prof. K. McIlhany,

Thank you for carefully reading the manuscript and providing helpful guidance for revisions. We highlighted the changes in the revised version of the manuscript using a blue font.

**Prof. K. McIlhany:** *I found the paper interesting and relevant.*

**Response:** Thank you for identifying the value of our work and providing a positive feed back.

**Prof. K. McIlhany:** *There are numerous grammatical errors however which need to be addressed. There are enough that I simply scanned my edited copy for your perusal. My strong suggestion is that one of the writers read aloud the paper so that these mistakes are caught early on before submitting in the future. I realize that some of the authors are non-native English speakers and I am always impressed at how well*

[Figure]

*they perform in the scientific environment, however, technical writing requires clarity. In some cases, the grammar issues lead to confusion about what exactly is being said.*

**Response:** We agree, our original version had too many spelling and grammar errors; hopefully our second try is not a hard slog. We have highlighted the corrections in blue in the revised version.

**Prof. K. McIlhany:** *On pages 4-5 there is some confusion in the text about whether the equations indicated boundary points as either "red" or "bold". Further, this confusion is present in table 1. The PDF file does not show "red" when it is indicated in table 1 but does show red in table 2. When printing the article in black and white, it is better to indicate these as bold, so perhaps they could be both red and bold and then everyone is happy.*

**Response:** Thank you for pointing out this issue. We have replaced the "red" letters from the "bold" letters to follow the journal standard.

**Prof. K. McIlhany:** *Also, on page 4 and 5 is repeated text, pick one and remove the other..*

**Response:** Thank you for pointing this out. We have removed the repeated text.

**Prof. K. McIlhany:** *Overall, I enjoyed the paper and its content. I do have a few technical questions, but nothing to prevent publication.*

**Response:** Thank you for your interest. We have addressed all the questions in the rest of this letter.

**Prof. K. McIlhany:** *1) Page 7: In seeking a minimum for the U-curve method, how many numerical attempts at alpha are reasonable?*

**Response:** In genaral, the range for the $\alpha$ varies from 0 to 1. In this work, we have used 100 of $\alpha$ values that varies from $10^{-12}$ to 1. We have added an explanation to clarify this in page 6 (modified version).

**Prof. K. McIlhany:** *2) Page 9: When the data contains a horizontal stripe, do you accidentally remove it (like a road) and if not, how do you know to avoid it?*

**Response:** This is a good question. And a valid criticism. We can break this question into two parts.

1. If there is a road, or any real signal that is exactly aligned with axis of the known stripes biased in a part of the image (assume that we have a part of the image with only biased data). In this case, we can destripe the image by defining the weight matrix $L$, considering a piece of image that does not have the horizontal road. We have added an explanation in page 3 for that.

2. If there is a road, or any real signal that is exactly aligned with axis of the known stripes biased in the complete image. Then they would be in danger of being regularized to disappear in a smooth "denoised" image and our only current protection agains this is the unlikeliness that a perfectly straight road would be both straight for long stretches, and furthermore straight and aligned with the sensor error. If this were deemed a general problem however, a mask to de-emphasize the regularization spatially could be developed in the regularity term at spatial locations where there is a known mapped road or other perfectly straight feature.

**Prof. K. McIlhany:** *3) Page 10: Why 12 for the threshold? How do you choose 12? A suggestion would be to turn Fig. 5 by 90 degs and then set the threshold to the FWHM of the highest peak found. You could repeat this process using the next lowest peak*

[Figure]

*and see how many more stripes are found. Eventually, you will see a rise in the number of stripes found and then raise the threshold to an earlier value when the number of stripes found was more consistent.*

**Response:** We take the suggestion and we have included an explanation for that in page 10.

**Prof. K. McIlhany:** *4) When citing "inpainting" did you consider Andrea Bertozzi?*

**Response:** We didn't consider Andrea Bertozzi for this work. We referred M. Bertalmio and we have cited the author.

**Prof. K. McIlhany:** *Most of my grammatical issues are highlighted in red ink on the attached PDF. I answered my own inquiries about the "$argmin_u$" and "vicarious" comments, so please disregard those, but the rest are genuine issues about writing.*

**Response:** We have answered all the other comments.

**Prof. K. McIlhany:** *Thank you for the opportunity to review your work*

**Response:** Thank you for the valuable comments which helped to improve the quality of our article.

**Supplement:**

[revised manuscript text omitted]

---

## Referee Comment (RC2) · Anonymous Referee #2 · 3 Apr 2017

The paper is overall well-written and describes how to deal with an important problem when using remote sensing data, especially for using infrared and visible frequencies for satellite imagery, that of striping. The paper presents a method which is able to diminish and correct the impact of striping.

Despite some minor grammar and orthographic errors, the paper is well-written, explains the problem clearly, presents the method in a clean manner and provides a sufficient amount of details of it. My only real concern with this paper is its suitability for Nonlinear processes in geophysics, as no nonlinear geophysical process is described in all the paper, just a processing technique (interesting as it is).

Some minor comments:

- How is the direction of stripes identified in general?

- What happens if the stripes contain valid information, i.e., there is an offset and/or a rescaling? Shouldn't they be consider, after readjustement?

- Eq. 9 has more undetermination that just a constant value: any function in the kernel of the operator $D_{xx}+\alpha L D_{yy}$ can be added to a solution and will yield a new solution. In fact, the point is that the matrix A is non-invertible. This is connected with the discussion on condition numbers in Section 2.3, but prior to go directly to discuss any regularization I think this point deserves some comments.

The issue is significant for instance on page 7, when developing the U-curve method, as one important parameter is the minimum non-zero singular value. How do you decide that some value is non-zero for a given numerical precision? A threshold is for sure used, and the point should be clarified, explaining in particular this choice.

Chorophyll images are not as smooth as claimed, chlorphill concentration being very intermitten. Even SST present strong frontal zones that break smoothness. Along fronts they are indeed smooth, but not across fronts, so anisotropy is a key ingredient. Some problems may arise with the parte of the front that is eventually aligned with the stripe direction. Please comment the issue.

The absolute percentage error on page 10 is not correctly defined, as referring to a value with a conventional origin is meaningless (imagine how this error would change is you take the SST in Kelvin or in Celsius, for instance). It is much more customary to compare errors to the dynamic range of the image (for instance, as measured by the standard deviation of the values).

12 is not a magic number; please be more descriptive about how to chosing the threshold in figure 5. And please provide units.

Although it is a bit beyond of the scope of the paper, it will be very convenient to have a in-situ validation dataset for verifiying if the destriped images are of higher quality.

---

## Author Response (AR1)

**Responses to the Comments made by Referee #2**

Ranil Basnayake, Erik Bollt, Nicholas Tufillaro, Jie Sun, and Michelle Gierach

April 29, 2017

We thank the reviewer for valuable comments. We highlighted the changes in the revised version of the manuscript using a blue font.

**Referee #2:** *The paper is overall well-written and describes how to deal with an important problem when using remote sensing data, especially for using infrarred and visible frequencies for satellite imagery, that of striping. The paper presents a method which is able to diminish and correct the impact of striping.*

**Response:** Thank you for providing positive comments about our work.

**Referee #2:** *Despite some minor grammar and orthographic errors, the paper is well-written, explains the problem clearly, presents the method in a clean manner and provides a sufficient amount of details of it.*

**Response:** We agree, our original version had grammar and orthographic errors. Hopefully the current version is free of those errors. We have highlighted the corrections in blue in the revised version.

**Referee #2:** *My only real concern with this paper is its suitability for Nonlinear processes in geophysics, as no nonlinear geophysical process is described in all the paper, just a processing technique (interesting as it is).*

**Response:** We considered NPG as a suitable outlet for our paper especially because of the journal's statement that "The editors encourage submissions that apply nonlinear analysis methods to both models and data." In this regard, we feel that our paper makes a good candidate for this journal. In addition, this paper concerns a data issue regarding remote sensing in the field of geoscience. Both referees have responded positively to this paper, and we believe the general readership will as well. Therefore it is our opinion that experimental issues related to the theme of the journal are a good topic for publication here, and we hope the editor and referees will agree with this point.

**Some minor comments:**

**Referee #2:** *How is the direction of stripes identified in general?*

**Response:** The stripes are due to details of the optical sensor on board the satellite camera. Therefore the exact alignment of this cause of the stripes is well known by the known orientation of the satellite.

**Referee #2:** *What happens if the stripes contain valid information, i.e., there is an offset and/or a rescaling? Shouldnt they be consider, after readjustement?*

**Response:** This a good question. If an actual stripe contains valid information, we are not able to do any readjustment in our destriping model. On the other hand, if the real data appears as a stripe, then the question can be broken into two parts.

1. If there is a road, or any real signal that is exactly aligned with axis of the known stripes biased in a part of the image (assume that we have a part of the image with only biased data). In this case, we can destripe the image by defining the weight matrix $L$, considering a piece of image that does not have the horizontal road. We have added an explanation in page 3 for that.

2. If there is a road, or any real signal that is exactly aligned with axis of the known stripes biased in the complete image. Then they would be in danger of being regularized to disappear in a smooth denoised image and our only current protection agains this is the unlikeliness that a perfectly straight road would be both straight for long stretches, and furthermore straight and aligned with the sensor error. If this were deemed a general problem however, a mask to de-emphasize the regularization spatially could be developed in the regularity term at spatial locations where there is a known mapped road or other perfectly straight feature.

**Referee #2:** *1) Eq. 9 has more undetermination that just a constant value: any function in the kernel of the operator $D_{xx} + \alpha L D_{yy}$ can be added to a solution and will yield a new solution. In fact, the point is that the matrix A is non-invertible. This is connected with the discussion on condition numbers in Section 2.3, but prior to go directly to discuss any regularization I think this point deserves some comments.*

**Response:** This is an excellent comment. We'd like to clarify that even though the matrix A appears non-invertible, with a non-empty kernel, in this work we have imposed "reflexive" boundary conditions parallel to the stripes and "zero" boundary conditions transverse to the stripes which makes A a full-rank matrix, thus leading to uniqueness of solution. This important issue is now explicitly discussed on page 4, (1) last sentence of the first paragraph, (2) in the paragraph immediate above Section 2.1.

**Referee #2:** *2) The issue is significant for instance on page 7, when developing the U-curve method, as one important parameter is the minimum non-zero singular value. How do you decide that some value is non-zero for a given numerical precision? A threshold is for sure used, and the point should be clarified, explaining in particular this choice.*

**Response:** Thank you for raising this question and it helped us to provide correct notations for the definition of $U$-curve method. We have changed the notations in Eq. (11), Eq. (12) and $5^{th}$ line in page 6. In this computation, we used $10^{-12}$ as the smallest value in the selection interval of $\alpha$. In this way, we can give the threshold for the smallest singular value to be larger than $10^{-18}$ as we can find an appropriate $\alpha \in \left( \sigma_r^{2/3}, \sigma_1^{2/3} \right)$.

**Referee #2:** *3) Chorophyll images are not as smooth as claimed, chlorophill concentration being very intermitten. Even SST present strong frontal zones that break smoothness. Along fronts they are indeed smooth, but not across fronts, so anisotropy is a key ingredient. Some problems may arise with the parte of the front that is eventually aligned with the stripe direction. Please comment the issue.*

**Response:** We agree that the images are not always smooth and hence the computation of $S$ curve may be affected by non-smoothness of the images. In this case, we should not pick the whole image to compute the $S$ curve, but an image segment that is smooth enough to identify the stripes from the $S$ curve. We have explained it in second paragraph of Section 3.2 (page 11-12).

**Referee #2:** *4) The absolute percentage error on page 10 is not correctly defined, as referring to a value with a conventional origin is meaningless (imagine how this error would change is you take the SST in Kelvin or in Celsius, for instance). It is much more customary to compare errors to the dynamic range of the image (for instance, as measured by the standard deviation of the values).*

**Response:** We agree that the "absolute percentage error" is not invariant. However, in this work, we try to define an error metric that can use to visualize the error between the striped and destriped images. When we work with real world data, we only can compare the data which are not on the stripes and this "absolute percentage error" shows us how the computations affect such data. Therefore, we believe that this dentition works well for the our purpose.

**Referee #2:** *12 is not a magic number; please be more descriptive about how to chosing the threshold in figure 5. And please provide units.*

**Response:** Thank for pointing out this and the correct number shoud be 18. We have included an explanation for that in page 10. Also thank you for reminding us to put the units for the threshold value and it will definetely help reader to understand the concept of the $S$ curve. We have added units with explanations at the $5^{th}$ line in page 10, Fig. 5, $2^{nd}$ line in page 12 and $5^{th}$ line in page 15.

**Referee #2:** *Although it is a bit beyond of the scope of the paper, it will be very convenient to have a in-situ validation dataset for verifiying if the destriped images are of higher quality.*

**Response:** Thank you for suggesting a data set for the validation and absolutely it is beyond of the scope of the paper. We will surely try it in our future work.

[revised manuscript text omitted]